# Mitochondria-Targeting Antioxidant Provides Cardioprotection through Regulation of Cytosolic and Mitochondrial Zn^2+^ Levels with Re-Distribution of Zn^2+^-Transporters in Aged Rat Cardiomyocytes

**DOI:** 10.3390/ijms20153783

**Published:** 2019-08-02

**Authors:** Yusuf Olgar, Erkan Tuncay, Belma Turan

**Affiliations:** Departments of Biophysics, Ankara University Faculty of Medicine, 06100 Ankara, Turkey

**Keywords:** aging-heart, intracellular free zinc, zinc-transporters, cardiovascular function, oxidative stress, insulin resistance, mitochondria

## Abstract

Aging is an important risk factor for cardiac dysfunction. Heart during aging exhibits a depressed mechanical activity, at least, through mitochondria-originated increases in ROS. Previously, we also have shown a close relationship between increased ROS and cellular intracellular free Zn^2+^ ([Zn^2+^]_i_) in cardiomyocytes under pathological conditions as well as the contribution of some re-expressed levels of Zn^2+^-transporters for redistribution of [Zn^2+^]_i_ among suborganelles. Therefore, we first examined the cellular (total) [Zn^2+^] and then determined the protein expression levels of Zn^2+^-transporters in freshly isolated ventricular cardiomyocytes from 24-month rat heart compared to those of 6-month rats. The [Zn^2+^]_i_ in the aged-cardiomyocytes was increased, at most, due to increased ZIP7 and ZnT8 with decreased levels of ZIP8 and ZnT7. To examine redistribution of the cellular [Zn^2+^]_i_ among suborganelles, such as Sarco/endoplasmic reticulum, S(E)R, and mitochondria ([Zn^2+^]_SER_ and [Zn^2+^]_Mit_), a cell model (with galactose) to mimic the aged-cell in rat ventricular cell line H9c2 was used and demonstrated that there were significant increases in [Zn^2+^]_Mit_ with decreases in [Zn^2+^]_SER_. In addition, the re-distribution of these Zn^2+^-transporters were markedly changed in mitochondria (increases in ZnT7 and ZnT8 with no changes in ZIP7 and ZIP8) and S(E)R (increase in ZIP7 and decrease in ZnT7 with no changes in both ZIP8 and ZnT8) both of them isolated from freshly isolated ventricular cardiomyocytes from aged-rats. Furthermore, we demonstrated that cellular levels of ROS, both total and mitochondrial lysine acetylation (K-Acetylation), and protein-thiol oxidation were significantly high in aged-cardiomyocytes from 24-month old rats. Using a mitochondrial-targeting antioxidant, MitoTEMPO (1 µM, 5-h incubation), we provided an important data associated with the role of mitochondrial-ROS production in the [Zn^2+^]_i_-dyshomeostasis of the ventricular cardiomyocytes from 24-month old rats. Overall, our present data, for the first time, demonstrated that a direct mitochondria-targeting antioxidant treatment can be a new therapeutic strategy during aging in the heart through a well-controlled [Zn^2+^] distribution among cytosol and suborganelles with altered expression levels of the Zn^2+^-transporters.

## 1. Introduction

Aging in humans is programmed genetically and modified by environmental influences as well as having a combination of morphological and functional changes in organs, tissues, and cells [1]. The rate of aged human percentage among populations is accelerating all over the world. The effect of aging can vary widely from one to another individual [2]. High resistance to blood pumping and increased workload against the increased resistance are the general findings in aged individuals [3]. With a short statement, human aging is a time-related relapse of the physiological functions during the life span, including insufficient cardiovascular functioning. Recent and even early both clinical and experimental studies mentioned that QT-intervals in electrocardiogram were prolonged with age in an overtly healthy older population [4,5,6,7]. Furthermore, there are a number of studies demonstrating a relationship between insulin resistance and cardiac dysfunction in elderly individuals and experimental animals [6,8,9,10]. Indeed, one component of aging-associated development of cardiac dysfunction is alteration in cardiac energy metabolism [11,12,13]. Moreover, an important amount of literature data documented that there is a close link, which shows the involvement of oxidative stress in the pathogenesis of aging. The production of an increasing amount of lipid peroxidation and its accumulation into the heart during aging can increase the susceptibility to the development of cardiovascular diseases besides others, at most, due to the production of reactive oxygen species (ROS) and free radicals [6,14,15]. More importantly, there are early studies demonstrating the role of zinc in biological samples, associated with dietary zinc and the levels of lipid peroxidation as well as protein thiol oxidation in the mammalian tissues [16,17].

At the cellular level, there are several damaging changes in cardiomyocytes such as marked increases in their dimensions, decreases in the number of living and functional cardiomyocytes [6]. Furthermore, in terms of cellular electrical activities, among other observations, there were significant action potential prolongation, changes in ionic fluxes across sarcolemma, and dysregulation in intracellular free Ca^2+^ level ([Ca^2+^]_i_), mitochondrial defects and increased oxidative stress in cardiomyocytes [11,18,19,20,21,22]. Although there is not yet a consensus on the role of mitochondria in aging-associated insufficient cardiomyocyte function, mitochondria-associated increases in intracellular ROS seems to play a central role during physiological mammalian aging. It is well accepted that the main role of mitochondria is to provide ATP demand and modulate the cytosolic Ca^2+^-signaling, influencing cellular ROS, and regulating the redox state of cells [23,24,25]. On the other hand, it has been also demonstrated that exposing oxidants could cause a very important increase in cytosolic free Zn^2+^ level in cardiomyocytes ([Zn^2+^]_i_), comparable to that of [Ca^2+^]_i_ [26]. Furthermore, we have recently shown that the high [Zn^2+^]_i_ could induce marked activation in ATP-sensitive K^+^-channel currents, depending on the cellular ATP levels [27], while a depletion in ATP production in insulin-resistant rat ventricular cardiomyocytes could also induce significant prolongation in action potentials [6,28]. All of these already known to imply a cross-correlation between insulin resistance, alterations in [Zn^2+^]_i_ and cardiac dysfunction in the aged rat cardiomyocytes. Moreover, there is also a strong link between mitochondrial metabolism, ROS generation, and the senescent state. In this regard, early studies demonstrated also that mitochondrial mutations increase in frequency with age in both animal models and in humans [29,30]. Furthermore, since a depolarization-induced Zn^2+^-uptake induced the high amount of cell death and enhanced accumulation of mitochondrial free Ca^2+^ ([Ca^2+^]_Mit_) in Ca^2+^- and Zn^2+^-containing media, the interactive roles of Zn^2+^ and Ca^2+^ in mitochondrial dysfunction have been studied and demonstrated in neural studies [31]. Moreover, these studies also showed how increased [Ca^2+^]_Mit_ led to important impairments in different intracellular signal transduction pathways, including cellular oxidative stress and apoptosis [2,32,33].

In general, zinc, being a multipurpose element for the mammalian body, plays important roles for the regulation of several cellular signaling mechanisms as Zn^2+^ [34]. Although there are limited numbers of data existing in the field of the role of [Zn^2+^]_i_ in the aged heart, few studies have demonstrated the role of alterations in [Zn^2+^]_i_-homeostasis, at least, via alterations in Zn^2+^-transporting proteins (ZIP and ZnT families). In this regard, it has been widely discussed the importance of mitochondrial Ca^2+^-homeostasis as potential target event for mitochondrial medicine [35,36].

The [Zn^2+^]_i_-homeostasis and the suborganelle levels of [Zn^2+^] are altered under pathological conditions, including hyperglycemia in cardiomyocytes [37,38]. Moreover, we have shown that a re-distribution of the expression level of some Zn^2+^-transporters can play an important contribution to those changes [37,39]. In the literature, there is a limited number of studies on the role of cellular [Zn^2+^] and Zn^2+^-transporters in the aged heart function [40]. In this field, authors demonstrated the fact of ZIP14 role as an inflammation responsive transporter and it has important phenotypic effects, amplified with aging [41]. Therefore, in the present study, we first aimed to examine the distribution of cellular [Zn^2+^] among suborganelles, such as S(E)R and mitochondria ([Zn^2+^]_SER_ and [Zn^2+^]_Mit_) in ventricular cardiomyocytes from aged rats compared to young ones (24-month vs. 6-month). Second, we examined a possible contribution of redistribution of Zn^2+^-transporters among suborganelles in these aged cardiomyocytes. Third, using a mitochondrial-targeting antioxidant MitoTEMPO, we examined the important contribution of mitochondria to the aging-associated changes into aged cardiomyocyte dysfunction.

## 2. Results

### 2.1. General Parameters of the Aged Rats

The average body weights of the aged and young rats were 380±12 g and 310±8 g, respectively. The heart to body weight ratio was higher in the 24-month old rats (aged group; number of rats: 10) than the 6-month old rats (young group; number of rats: 9), significantly (5.5±0.3 mg/g vs. 4.2±0.1 mg/g), indicating the slightly but significantly development of hypertrophy. The fasting blood glucose level was also slightly but significantly high in the old rats compared to the young rats (93±3 mg/dL vs. 82±2 mg/dL). As shown previously, there was a significant presence of insulin resistance in the old group with the measurement of oral glucose tolerance test (OGTT) and homeostasis model assesment of of insulin resistance (HOMA-IR) index, as described previously [6]. Although the aged group has a similar fasting blood glucose level with a young group, they have about a 17% high blood glucose level during 30-min of OGTT and that increased level during 60, 90, and 120 min measurements was still high, by about 10% compared to those of the young group. The HOMA-IR index was measured as 7.8±1.3 in the aged groups and 6.1±1.0 in the young group, in which the differences between these two groups were significantly different.

### 2.2. MitoTEMPO Regulation of [Zn^2+^]_i_ and Some Zn^2+^-Transporters in the Aged Left Ventricular Cardiomyocytes

It has been already demonstrated that there is a close relationship between increases in oxidative stress, at most, ROS and [Zn^2+^]_i_ in cardiomyocytes under pathological condition [26,37,39], while the marked increase in ROS was demonstrated in the aged cardiomyocytes. Therefore, here, we first determined the cellular [Zn^2+^]_i_ in the aged ventricular cardiomyocytes isolated from 24-month old rats. As can be seen in Figure 1A,B, the average level of [Zn^2+^]_i_ was significantly high (about 3-fold) in the aged cardiomyocytes loaded with FluoZin-3 compared to those of 6-month old rats. A mitochondria-targeting antioxidant MitoTEMPO treatment (1-µM for 5-h) of the cells provided marked recovery in this high [Zn^2+^]_i_ in the aged cardiomyocytes.

To examine the regulation of some Zn^2+^-transporters such as ZIP7, ZIP8, ZnT7, and ZnT8, which were previously investigated in the ventricular cardiomyocytes under pathological conditions [37,39], we determined their protein expression levels in total cell homogenates using Western blot analysis. As shown in Figure 1C,D, there were significantly increased ZIP7 (left in C) with decreased ZIP8 (right in C), whereas the ZnT7 level was found to decrease significantly (left in D) with increased ZnT8 (right in D). These changes could recover significantly a treatment of the aged cells with mitochondrial antioxidant MitoTEMPO.

### 2.3. Aging-Associated Redistribution of Cellular [Zn^2+^]_i_ in the Aged Cardiomyocytes

To examine the redistribution of the cellular [Zn^2+^]_i_, we used a cell model to mimic the aged cell by using rat ventricular cell line H9c2, as used previously [37,39,42]. As can be seen in Figure 2A,B, we determined the [Zn^2+^]_i_ level in H9c2 cells with galactose treatment (D-gal, 50-µM, about 3-fold higher than that of nontreated cells) similar to one obtained in ventricular cardiomyocytes isolated freshly from 24-month old rats (about 3-fold higher than that of 6-month old rats given in Figure 1B).

Following the validation of the cellular [Zn^2+^]_i_ in these D-GAL50 treated cells, we determined [Zn^2+^] levels in both mitochondria and S(E)R, separately ([Zn^2+^]_Mit_ and [ Zn^2+^]_SER_). As can be seen in Figure 2C,D, respectively, the [Zn^2+^]_Mit_ increased over 2-fold in the aged cardiomyocytes compared to those of 6-month old rats. However, although there was a slight decrease in [Zn^2+^]_SER_ level measured in the aged group, the differences between the aged group and young group were not statistically significant. Furthermore, MitoTEMPO treatment (1-µM for 24-h) fully preserved the increased level of [Zn^2+^]_Mit_ in the aged cardiomyocytes (Figure 2C).

### 2.4. Confirmation of Mitochondrial Function and ROS Level in Aging-Modeled H9c2 Cells

To validate the aging in D-GAL50 treated H9c2 cells, we measured the mitochondrial membrane potential (MMP) and ROS level in these cells. As can be seen in Figure 3A–D, respectively, MMP was found to be depolarized and the ROS level was significantly high in D-GAL50 treated H9c2 cells compared to those of untreated H9c2 cells.

We further treated these D-GaL50 treated H9c2 cells with MitoTEMPO (1-µM for 24-h incubation) and then measured the MMP and ROS level. These parameters were found to be fully preserved with MitoTEMPO treatment.

### 2.5. Redistribution of Zn^2+^-Transporters among Suborganelles in the Aged Cardiomyocytes İsolated from the Left Ventricle of 24-Month Old Rats

We first isolated either mitochondria or S(E)R and cytosolic fractions in the isolated left ventricular cardiomyocytes (Figure 4A, Figure 5A, respectively). Following the validation of these fractions with specific markers (COXIV for mitochondria and SERCA2a for S(E)R), we determined the protein expression levels of the Zn^2+^-transporters given in Section 2.2. The evaluation of these transporters in isolated S(E)R fraction showed that both ZIP7 (B) and ZIP8 (C) did not change in the mitochondria fraction of the aged cardiomyocytes compared to those of young cardiomyocytes. However, both ZnT7 and ZnT8 did increase in these fractions significantly (D and E, respectively). MitoTEMPO treatment (1-µM, for 5-h) of the aged cardiomyocytes fully preserved the changes in the expression levels of these Zn^2+^-transporters measured in the mitochondria fraction.

The ZIP7 level (with respect to the SERCA2a level) was increased significantly in S(E)R fraction of the aged cardiomyocytes isolated from the 24-month old rats compared to those of 6-month old rats (Figure 5B) with no change in ZIP8 (C). Furthermore, the level of ZnT7 was markedly depressed in the S(E)R fraction of these aged cells compared to those of young cells (D) with no change in ZnT8 level (E). Furthermore, MitoTEMPO treatment the aged cardiomyocytes fully preserved the changes in the expression levels of these Zn^2+^-transporters measured in the S(E)R fraction.

### 2.6. MitoTEMPO Preserves the Increases in Both Oxidation and K-Acetylation Levels in Left Ventricular Aged Cardiomyocytes

In order to detect the direct effect of aging on the heart via changes in oxidation status of the cells, we first measured the levels of total oxidant status (TOS) and total antioxidant status (TAS) in the aged cardiomyocytes compared to the young cardiomyocytes. Figure 6A shows that the TOS level (left) was markedly high in the aged cardiomyocytes compared to those of young group, while the TAS level (right) in the age group was significantly lower than those of the young group. MitoTEMPO treatment (1-µM for 5-h) of the aged cardiomyocytes not fully but significantly preserved these changes.

To demonstrate the effects of aging on the protein-thiol oxidation level in isolated left ventricular cardiomyocytes, we measured the total and oxidized protein-thiol levels in the aged cardiomyocytes compared to those of young cardiomyocytes. As can be seen in Figure 6B, aging induced significant increases in oxidized protein-thiol level (right) with no change in total protein-thiol level (left). MitoTEMPO treatment of the aged cardiomyocytes fully preserved these changes measured in the aged-group.

In the last part of our study, we determined the total K-Acetylation level in isolated left ventricular aged cardiomyocytes. Compared to those of the young group, the total K-Acetylation was significantly high in the aged group (Figure 6C). Further evaluation of the K-Acetylation level in isolated suborganelle fractions, the mitochondrial K-Acetylation (Figure 6D) was markedly high in the aged cardiomyocytes compared to those of young cardiomyocytes, whereas there was no change in the S(E)R fraction K-Acetylation (Figure 6E). MitoTEMPO (1-µM for 5-h) treatment of the aged cardiomyocytes fully preserved these changes to the levels of the young cardiomyocytes.

## 3. Discussion

The overall aim of the present study was first to examine the distribution of cellular [Zn^2+^] among suborganelles, such as S(E)R and mitochondria ([Zn^2+^]_SER_ and [Zn^2+^]_Mit_) in ventricular cardiomyocytes during aging in mammalians. Second, we examined whether some Zn^2+^-transporters can redistribute among suborganelles and contribute to redistributed of free Zn^2+^ among suborganelles in these aged cardiomyocytes. Third, to understand whether [Zn^2+^]_Mit_ plays a predominant role to the alteration of aging-associated alterations in cardiac function by using a mitochondrial-targeting antioxidant, MitoTEMPO. Similar to previous data obtained in hyperglycemic cardiomyocytes [39,43], we monitored and quantified significantly high [Zn^2+^]_i_ in aged cardiomyocytes compared to those of young cells by using confocal examinations. Furthermore, our present data also demonstrated that there is a redistribution of cellular [Zn^2+^]_i_ among suborganelles under aging, ones similar to those of hyperglycemic cardiomyocytes [37,39]. These altered levels of either cellular [Zn^2+^]_i_ or redistribution of cellular compartments can likely be one of the potential mediators for the age-dependent development of cardiac dysfunction. Furthermore, our present results suggest that these Zn^2+^-carriers, such as ZIP7 and ZnT7 as well as ZIP8 and ZnT8, can carry Zn^2+^ in opposite directions across either S(E)R or mitochondria membranes, which are similar to ones previously demonstrated in hyperglycemic cardiomyocytes [37,39]. In addition, the activity of these transporters depends on modulations oxidation, phosphorylation, or K-Acetylation during aging in cardiomyocytes, and, therefore, these changes in those carries, in turn, contribute to that of redistribution of cellular [Zn^2+^]_i_ among compartments. In this regard, we previously have demonstrated that high cellular [Zn^2+^]_i_ could contribute to both electrical and mechanical dysfunction in left ventricular cardiomyocytes, in a manner of a close relationship between increased ROS and depressed cellular activity through high cellular [Zn^2+^]_i_ [39,43]. Importantly, it has been considered that [Zn^2+^]_i_ increases resulted in releases of Zn^2+^ during cardiac-cycle [44], which can further trigger the production of higher pro-oxidants, leading to more oxidative damage in ventricular cardiomyocytes [43]. With a reversed relationship, oxidants in the cells can induce significant increases in [Zn^2+^]_i_, in part, the release of Zn^2+^ from metalloproteins [26], while we observed marked alterations in the heart function in hyperglycemic cardiomyocytes through increased oxidative stress and high [Zn^2+^]_i_ [45]. Evidently, it has been pointed out that not only [Zn^2+^]_i_ but also [Ca^2+^]_i_ was increased in left ventricular cardiomyocytes under oxidative stress to different extents [26]. Another important point related with increases in [Zn^2+^]_i_ (about 70% with much less increase in [Ca^2+^]_i_ in cardiomyocytes) under increased oxidative stress in mammalians such as diabetic rats, the parallelism between ionic changes and an unbalanced oxidant-status/antioxidant capacity [46]. More importantly, supporting our present data, authors demonstrated that elevations in [Ca^2+^]_i_ and [Ca^2+^]_mit_ were coupled with [Zn^2+^]_i_ and [Zn^2+^]_mit_ in cardiomyocytes under aldosteronism [47].

Zn^2+^ is a redox-inert ion in biology, and, therefore, it also plays important role in redox-status of cells while its levels are modulated by the redox-state of the cells such as ROS production [48], although Zn^2+^ has opposing effects in the cells including either elevate the antioxidant capacity or lead to ROS release at toxic levels [49]. Indeed, literature data documented that zinc-coordination with cysteine ligands can induce the sulfur-ligands oxidization [46,49], while there is a relationship between increased [Zn^2+^]_i_ and elevation of ROS in mammalian cells by inhibiting mitochondria [50,51]. Since it has been shown that oxidative phosphorylation rates and and ATPase-activities were markedly depressed in mitochondria from heart preparations under pathological conditions, including hyperglycemia [52], our similar findings in the present study strongly support the current hypothesis related with the important role of mitochondrial function via changes not only in [Zn^2+^]_i_ but also in [Zn^2+^]_Mit_ in heart dysfunction during aging with insulin resistance development. In addition, the high protein thiol oxidation and depressed antioxidant capacity in the aged cells (although aging is a physiological event) are in accordance with our previous finding with hyperglycemic cardiomyocytes. Indeed, we did present previously that the increased [Zn^2+^]_i_ was associated with high phosphorylation in cardiac Ca^2+^–release channels (RyR2) via phosphorylation of both protein kinase A (PKA) and CamKII [43]. In summary, it seems that the increased [Zn^2+^]_mit_, with an important contribution to increases in both [Zn^2+^]_i_ and [Ca^2+^]i, under increased oxidative stress together with depressed antioxidant-defense in the cells under aging, can induce important alterations in cardiomyocyte function. Therefore, one can suggest that a disbalance in [Zn^2+^]_i_ in cardiomyocytes, under either pathological conditions or aging, can result in a disbalance in cellular signaling mechanisms via interference between the [Zn^2+^]_i_ signaling and the [Ca^2+^]_i_ signaling of cardiomyocytes.

In the present study, we demonstrated that the total K-Acetylation measured in total cellular preparation was significantly high in the aged group compared to that of the young group. Furthermore, with a detailed examination in isolated organelle levels, the K-Acetylation level in isolated mitochondria fraction was significantly high, one similar to that of total cellular total acetylation; however, there were no differences in the S(E)R fraction K-Acetylation of these two groups. As known from previous studies, an altered mitochondrial function is an underlying basis for the increased sensitivity to oxidative stress in the aged heart, such as a decreased capacity to oxidize fatty acids and enhanced dependence on glucose metabolism [53,54]. Aging can impair the structure, and function of mitochondria and induces mitochondrial oxidative phosphorylation [6,54]. Our data are in line with what is already known in this field because protein K-Acetylation underlies the aging-associated cardiac diseases. Further data on increased ROS in the aged cardiomyocytes support the hypothesis on ROS-associated increase in protein acetylation [54]. Indeed, the authors showed a 58% increase in protein acetylation of mitochondria from 24-month old rat heart compared to those of 6-month old rats, while a ROS scavenger preserved all those changes.

Previously, we demonstrated that the expression levels of ZIP7 and ZnT7 in left ventricular cardiomyocytes under hyperglycemia acted as mediators of ER stress as well as ER stress-associated cardiac dysfunction [37]. These two transporters are carrying Zn^2+^ in opposite directions across S(E)R membranes and they can phosphorylate under pathological conditions, though likely inducing important changes in S(E)R with deleterious consequences for cell function [37]. Although the levels of ZIP7 and ZnT7 in the S(E)R fraction of aged cardiomyocytes changed similar to those of the hyperglycemic cardiomyocytes [37], here, the change in [Zn^2+^]_i_ was not statistically significant between aged- and young-group cardiomyocytes. In addition, we, with a further study, did show that both ZIP7 and ZnT7 localized to both mitochondria and S(E)R and contribute to cellular Zn^2+^-muffling between cellular-compartments in either hyperglycemic or hypertrophic cardiomyocytes via affecting S(E)R-mitochondria coupling [39]. Supporting these previous data, here, we provided further information that mitochondria behave as essential mediators of the regulation of cellular during aging and provide an important contribution to aging-associated insufficient cardiac function and/or heart dysfunction.

Our present data are supported with the previous data and with an already existing hypothesis for the presence of the same transporters localized to both S(E)R and mitochondria as well as to other compartments in the same cell being responsible from cellular [Zn^2+^] distribution in the cytosol and other organelles under physiological conditions [55]. Indeed, a differential cytosolic-localization of ZnT8 in β-cells has also previously been shown [56] in addition to a differential subcellular localization of the splice variants of ZnT5 in human intestinal cells [57]. Importantly, in the present study, we demonstrated that the mitochondrial levels of both ZnT7 and ZnT8 in the aged left ventricular cardiomyocytes were found to be markedly increased while ZIP7 and ZIP8 remainede unchanged, inducing a marked increase in [Zn^2+^]_Mit_. These increases can further affect not only S(E)R-mitochondria coupling but also the increase of ROS production into cells. These changes further can contribute to the disruption of mitochondrial dynamism during the aging heart. Our present data with MitoTEMPO, importantly, provided very important information about the role of mitochondria as a target compartment of cardiomyocytes to the aging-associated alterations at the cellular level. Our overall data, therefore, showed, for the first time, that mitochondria are an important target for aging and mitochondria directed antioxidants like MitoTEMPO can provide substantially an important cardio-protective benefit to control of cellular and subcellular [Zn^2+^] in the aged cardiomyocytes via altered levels of some Zn^2+^-transporters (Figure 7). Any therapy with mitochondria-targeted antioxidant will have an important impact as the cardioprotective approach for aging-associated cardiac dysfunction in mammalians.

## 4. Material and Methods

### 4.1. Experimental Animals

Experiments were performed with 24-month old (24-month, aged group) and 6-month old (6-month, young group) Wistar male rats. Rats were kept in standard animal housing rooms and they had tap water and fed with standard chow ad libitum, freely and daily. The fasting blood glucose level and oral glucose tolerance test was applied, as described previously [58]. All experimental procedures were performed in accordance with the standards of the European Community guidelines on the care and use of laboratory animals, and were approved by the Ankara University with a reference number of 2016-18-165 in accordance with the guide for the care and use of laboratory animals (dated: 21.09.2016).

### 4.2. Fresh Cardiomyocyte İsolation from the Left Ventricle

The hearts of all experimental animals, following being anesthetized by sodium pentobarbital (30 mg/kg, i.p.), were perfused retrogradely and performed standard enzymatic digestion, as described elsewhere [59]. Briefly, the cannulated hearts through the coronary arteries on a Langendorff apparatus were perfused (3–5 min) with a Ca^2+^-free solution (in mM): 145 NaCl, 5 KCl, 1.2 MgSO_4_, 1.4 Na_2_HPO_4_, 0.4 NaH_2_PO_4_, 5 2-[4-(2-hydroxyethyl)piperazin-1-yl]ethanesulfonic acid (HEPES), and 10 glucose at pH 7.4, bubbled with O_2_ at 37 °C and then followed the perfusion with a solution containing 1 mg/mL collagenase (Collagenase A, Boehringer) for 30–35 min. Left ventricular cardiomyocytes were isolated following the digestion with and the only the cells with rod-shaped, quiescent, and responses electric-field stimulation were used for all experiments The percentage of viable cells was 70–80% in every heart following increases of Ca^2+^ in the medium to a concentration of 1-mM.

To examine the role of a mitochondrial-targeting antioxidant on aged-cardiomyocytes, half of isolated cardiomyocytes from the 24-month old group was treated with MitoTEMPO (1-μM in the medium that cells were incubated) for 5-h before the experiments.

### 4.3. Cell Culturing

The H9c2 cell-line was derived from embryonic rat heart and purchased from the American Type Culture Collection (CRL 1446) and grown at 37 °C in 5% CO_2_ in Dulbecco’s modified Eagle’s medium (DMEM) in the presence of 5.5 mM glucose, as described previously [60]. Briefly, the cells were grown to a density of about 10^5^ cells/cm^2^ and cultured as a monolayer in supplemented with 10% fetal calf serum, 50 U/mL penicillin-G and 50 μg/mL streptomycin in a humidified atmosphere at 37 °C.

To obtain aged cells, the sub-confluent cells were treated with different concentration of reducing sugar D-galactose (D-GaL: 10-, 50-, and 100-mg/L) for 48-h to accelerate the glycation of macromolecules. Aging cells evaluated with a marked increase of cytosolic reactive oxygen species (ROS) with depolarized mitochondrial membrane potential (MMP) after the treatment. A group of D-Gal induced aged H9c2 cells were treated with 1-μM MitoTEMPO for 24-h.

### 4.4. Determination of Intracellular and Suborganelle Levels of Free Zn^2+^ by Confocal Microscopy Measurements

Intracellular (cytosolic) free Zn^2+^ level ([Zn^2+^]_i_) was measured in freshly isolated cardiomyocytes by using confocal microscopy (Leica TCS SP5, Wuerzburg, Germany), as described previously [39]. To monitor Sarco(endo)plasmic reticulum and mitochondrial free Zn^2+^ levels ([Zn^2+^]_SER_ and [Zn^2+^]_Mit_), we used eCALWY sensors ER-eCALWY6 and Mit-eCALWY4, respectively. Image analysis was performed with ImageJ software (1.44p) using a homemade-macro, as described elsewhere [42]. To calculate free Zn^2+^, the maximum and minimum fluorescence ratios, obtained with either a heavy-metal-chelator *N*,*N*,*N*’, *N*’-tetrakis(2-pyridylmethyl) ethylenediamine, TPEN (50-μM) or Zn^2+^-saturation with 100-μM ZnCl_2_ and Zn^2+^-ionophore pyrithione (Zn^2+^/Pyr, 5-μM) and calculated with their specific Kd values (ER-eCALWY6: *K*_d_ = 2.9 nM; Mito-eCALWY4: *K*_d_ = 60 pM) for all groups. A group of D-Gal induced aged H9c2 cells were treated with 1-μM MitoTEMPO for 24-h.

### 4.5. Confocal Imaging of Mitochondrial Membrane Potential and ROS Level

The mitochondrial membrane potential, MMP, was measured in H9c2 cells using the fluorescence-based method similar to ones measured in freshly isolated cardiomyocytes, as described previously [61]. Briefly, cells were loaded with a membrane-permeant single wavelength fluorescence dye JC-1 (5-μM for 30-min) and imaged with a confocal fluorescence microscope (Leica TCS SP5). For calibration, carbonyl cyanide 4-(trifluoromethoxy)phenylhydrazone (FCCP; 5-μM) was used.

Confocal imaging of ROS levels in H9c2 cells, similar to ones in freshly isolated cardiomyocytes as described previously [61], was performed and quantified. Briefly, cells were loaded with a ROS indicator chloromethyl-2′,7′-dichlorodihydrofluoroscein diacetate (DCFDA, 10-µM for 60-min) and then are examined with a laser scanning microscope (Leica TCS SP5). For maximal fluorescence intensity, cells were exposed to a HEPES-buffered solution supplemented with H_2_O_2_ (100-μM).

### 4.6. Isolation of Sarco(endo)Plasmic Reticulum and Mitochondria Fractions from Left Ventricular Cardiomyocytes

The S(E)R-fractionation from isolated left ventricular cardiomyocytes was determined by using ER-isolation kit (Sigma, E0100), as described previously [37]. Briefly, cardiomyocytes were homogenized in isotonic extraction-buffer and then crude microsomal-fraction was isolated from post-mitochondrial fraction using ultracentrifugation. Western blot analysis was carried out using primary antibodies against ZIP7, ZIP8, ZnT7, and ZnT8. To confirm a proper S(E)R-isolation, SERCA2 (Santa Cruz, SC-8094) was used as S(E)R marker.

Isolation of mitochondrial fraction was performed using a mitochondria isolation kit (Thermo, 89874, Paisley, UK) in isolated cardiomyocytes, according to the manufacturer’s instructions, as described previously [39]. Protein levels were measured by Bradford Assay (Pierce Biotechnology, Waltham, MA, USA) with bovine serum albumin as standard solution, and homogenates were loaded on to 12% sodium dodecyl sulphate (SDS) polyacrylamide gels. To confirm a proper mitochondria-isolation, COX IV (Abcam, ab1474, Cambridge, UK) was used as a mitochondrial marker.

### 4.7. Western Blotting

Following incubations, cells were lysed in ice-cold Radioimmunoprecipitation assay buffer (RIPA, 50 μM Tris/HCl, pH at 7.4, 150 μM NaCl, 1% NP-40, 0.5% deoxycholate, 0.1% sodium dodecyl sulfate (*w*/*v*) and %1 protease inhibitor cocktail) by vortexing for 1 min. Following lysis, the samples were centrifuged at 10,000× rpm for 5-min and the protein content of the supernatants was determined using Bradford Assay (Pierce Biotechnology, Waltham, MA, USA) with bovine serum albumin as standard. Homogenates were loaded unto 8–10% SDS polyacrylamide gels and electrophoresed under constant voltage (150 V) for 2–3 h, and then electrically transferred onto polyvinylidene fluoride (PVDF) membrane for 30 min in the semi-dry transfer system. The membrane was blocked in 3–5% (*wt*/*vol*) albumin with 0.02% Tween-20 TBS buffer for 1 h and then incubated with recommended dilutions overnight with the primary antibodies; ZIP7 (Santa Cruz, sc-83858, 1:500, Dallas, TX, USA), ZIP8 (Protein Tech, 20459-1-AP, Manchester, UK) ZnT7 (Santa Cruz, sc-160948), ZnT8 (Santa Cruz, sc-98243), GAPDH (Santa Cruz, sc-365062), SERCA2 (Santa Cruz, sc-8094) purchased from Santa Cruz Biotechnology and COX4 (ab1474), from Abcam. The membranes were incubated with horseradish peroxidase-linked secondary antibodies and immunoreactivity was visualized using luminol-based chemiluminescence reaction. Immunoreactive bands on X-ray film were analyzed using open access ImageJ software (NIH) program.

All reagents were obtained from Sigma Aldrich (St. Louis, MO, USA) unless otherwise stated. All fluorescent dyes were purchased from Molecular Probes (Eugene, Oregon, USA). *N*,*N*,*N*′, *N*′-tetrakis-(2-pyridylmethyl)ethylenediamine (TPEN) were obtained from Sigma (St. Louis, MO, USA). TPEN was prepared as a 1-mM stock solution in dimethyl sulfoxide (DMSO).

### 4.8. Total Antioxidant Status (TAS) and Total Oxidant Status (TOS) Measurement in Cardiomyocytes

TOS and TAS levels were measured in plasma as well as in cardiomyocytes by using the commercially available kit (RL0024, *Rel Assay Diagnostics*, Gaziantep, Turkey), as described previously [62]. Briefly, we used the novel automated method, which is based on the bleaching of the characteristic color of a more stable ABTS (2,2′-Azino-bis(3-ethylbenzothiazoline-6-sulfonic acid)) radical cation by antioxidants. The results are expressed as mmol Trolox equivalent/L. The TOS level in plasma was measured using commercially available kits (RL0024, Rel Assay Diagnostics, Gaziantep, Turkey) as described previously [62]. Shortly, the oxidation reaction is enhanced by glycerol molecules abundantly present in the reaction medium. The ferric ion produced a colored complex with xylenol orange in an acidic medium. The color intensity, measured spectrophotometrically, is related to the total amount of oxidant molecules present in the sample. The assay is calibrated with H_2_O_2_ and the results are expressed in terms of µM H_2_O_2_ equivalent/L.

### 4.9. Determination of Oxidized Protein Thiol Level in Isolated Cardiomyocytes

Total and oxidized sulfhydryl (thiol) groups were estimated with Ellman’s reagent, as described previously [27]. Briefly, cardiomyocytes were thawed and lysed in 0.2 M Tris/HCl buffer pH at 8.1 containing 2% Na-dodecyl sulfate. For the measurement of total thiol-groups in cell homogenates, both for reaction mix and background reaction mix, cell lysate and 20% trichloroacetic acid mixtures were centrifuged for 10-min at 13,000 rpm. Following the centrifugation, the supernatants were collected and added NaOH. Both for total and oxidized thiol-group measurements, 2-mM 5,5′-dithiobis-(2-nitrobenzoic acid) was added to reaction mixtures and let them to stand for 20-min. Then, all reaction mixtures and background reaction mixtures were put in 96-well plates and absorbances were read using microplate-reader (SpectraMax Plus384, Molecular Devices LLC, San Jose, CA, USA) at 412-nm.

### 4.10. Determination of Protein K-Acetylation in Cardiomyocytes

Acetylation of lysine residues within proteins has emerged as an important mechanism used by cells to overcome many physiological processes. Assessment of lysine acetylation of proteins was performed according to the immune-blotting assay. As described above, cells were lysed, centrifuged and the protein content of the supernatants was determined [39]. Homogenates were electrophoresed and transferred onto polyvinylidene fluoride (PVDF) membrane in the semi-dry transfer system. The membrane was incubated with recommended dilutions overnight with the Anti-Acetyl lysin antibody (Abcam; ab21623). This antibody detects several bands including 11 kDa, 15 kDa, 45 kDa, and 50 kDa sizes. We analyzed a range between 30- and 50 kDa band sizes to correlate K-Acetylation of Zn-transporters of experimental groups.

### 4.11. Data Analysis and Statistics

All data were presented as a mean (±SEM) and a two-tailed Student’s *t*-test was used to determine statistical significance (*p* < 0.05).

## 5. Conclusions

In the present study, we have, for the first time, demonstrated that [Zn^2+^]_i_-homeostasis and its suborganelle levels such as [Zn^2+^]_Mit_ and [Zn^2+^]_SER_, are altered in left ventricular cardiomyocytes from 24-month old rats compared with those of 6-month old rats. Furthermore, we also demonstrated that these above alterations are due to not only changes in the total protein expression levels of the Zn^2+^-transporters such as ZIP7, ZIP8, ZnT7, and ZnT8 but also due to the re-distribution of these Zn^2+^-transporters in mitochondria and S(E)R. Moreover, our data, for the first time, have shown that the cellular changes in the parameters of ROS, both total and mitochondrial lysine acetylation, and protein-thiol oxidation are contributing to the changes in cellular [Zn^2+^]_i_-homeostasis in the aged cardiomyocytes. More importantly, confirming our hypothesis, a mitochondrial-targeting antioxidant, MitoTEMPO provided important cardioprotection via positively affecting the [Zn^2+^]_i_-dyshomeostasis of the ventricular cardiomyocytes from 24-month old rats.

## Figures and Tables

**Figure 1 ijms-20-03783-f001:**
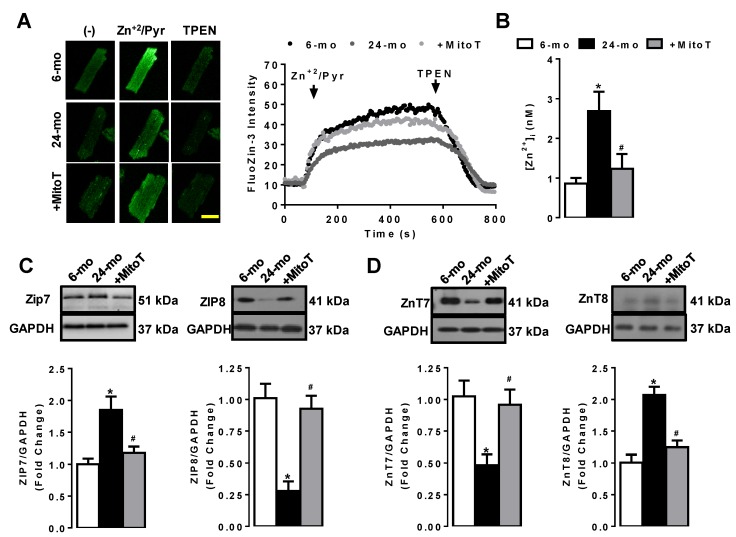
Determination of intracellular free zinc ([Zn^2+^]_i_) levels and associated Zn^2+^ transporters in the aged left ventricular cardiomyocytes. (**A**) Representing confocal images of cardiomyocytes and [Zn^2+^]_i_ measurement protocol in freshly isolated cardiomyocytes loaded with Zn^2+^-selective fluorescent dye FluoZin-3. Arrows on the right represent the Zn^2+^/Pyr and TPEN applications, respectively. The maximum and minimum fluorescent signals obtained, as described in the methods with Zn^2+^/Pyr (5-μM) and TPEN (50-μM). (**B**) The bars representing the average [Zn^2+^]_i_ for all groups. The protein expression levels of ZIP7 and ZIP8 (**C**, left and right, respectively), and for ZnT7 and ZnT8 (**D**, left and right, respectively) in total cellular homogenates. Cardiomyocytes were incubated with MitoTEMPO (1 µm for 5 h). Scale bar in the confocal image: 20 µm. Numbers of used cardiomyocytes isolated from 5–6 rats/groups were >20 per fluorescence protocols. Data were presented as Mean±SEM. The protein expression data were obtained with double assays in each sample from each group for each type of measurement (numbers of rats/groups = 5–6) and Significance level accepted as * *p* < 0.05 vs. 6-month old group, ^#^
*p* < 0.05 vs. 24-month old group.

**Figure 2 ijms-20-03783-f002:**
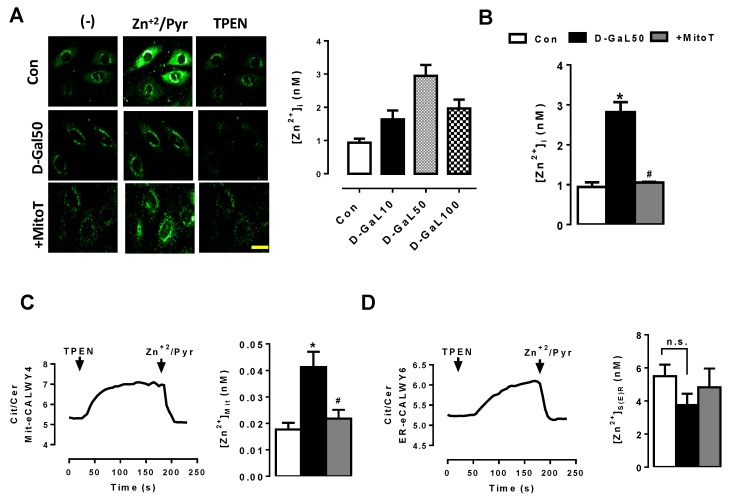
Determination of intracellular and subcellular [Zn^2+^]_i_ levels in D-galactose induced aged H9c2 cell line. (**A**) representative confocal images of H9c2 cells incubated with reducing sugar D galactose (D-Gal, 50 mg/mL for 48-h incubation) to monitor [Zn^2+^]_I_; (**B**) the measured [Zn^2+^]_i_ values in H9c2 cells with different concentrations of (D-Gal (10-,50-,100 mg/mL for 48-h) (left). Effects of mitochondria-targeted antioxidant MitoTEMPO treatment, (MitoT; 1 µM for 24-h) on [Zn^2+^]_i_ in D-Gal (50 mg/L) treated H9c2 cells (right); (**C**) representing graph to measure the [Zn^2+^]_Mit_ in a genetically encoded FRET-based Zn^2+^ sensor (Mit-eCALWY4) in D-Gal treated H9c2 cells (left). The measurement protocols are given in methods. The bar graph is representing the effect of the MitoTEMPO treatment on [Zn^2+^]_mit_ level; (**D**) representing graph to measure the level of Zn^2+^ in Sarco(endo)plasmic reticulum [Zn^2+^]_S(E)R_ in a genetically encoded FRET-based Zn^2+^ sensor (ER-eCALWY6) (left) and effect of MitoTEMPO treatment on [Zn^2+^]_SER_ (right). Arrows (in C and D) represent the TPEN and Zn^2+^/Pyr applications, respectively. Numbers of used cardiomyocytes isolated from 5–6 rats/groups were >25 per fluorescence protocols. Scale bar: 10-µm. Data were presented as Mean±SEM. Data were presented as Mean±SEM. Significance level accepted as * *p* < 0.05 vs. untreated H9c2 group, ^#^
*p* < 0.05 vs. D-Gal treated H9c2 group.

**Figure 3 ijms-20-03783-f003:**
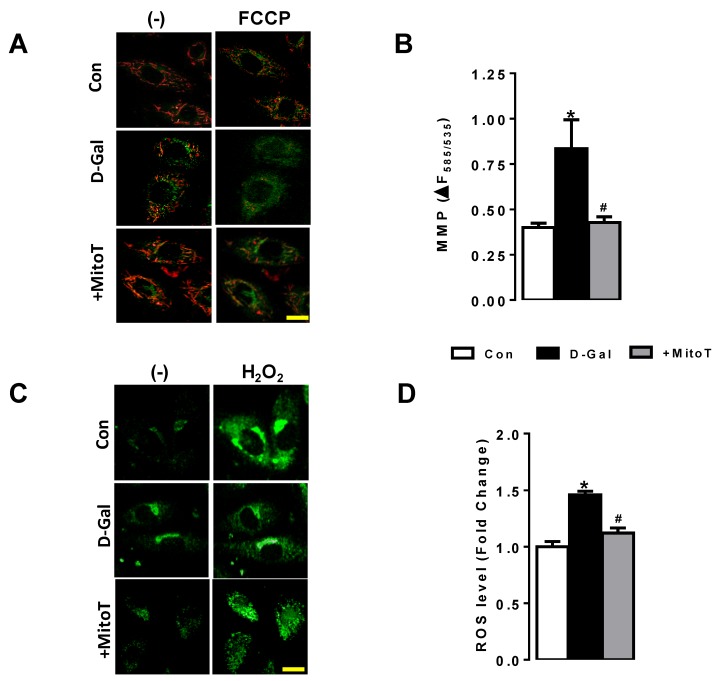
Validation of increases in Reactive Oxygen Species (ROS) and depolarization in Mitochondrial Membrane Potential (MMP) in D-Gal induced H9c2 cells (aging mimicked cells). (**A**) representative confocal images of JC-1 (5-μM, 30-min) loaded H9c2 cells. The probes were excited at 488 nm, and the red fluorescence image was detected at both 535 and 585 nm. To calibrate the changes in MMP, cyanide 4-(trifluoromethoxy)phenylhydrazone (FCCP, 5-μM) was used; (**B**) MitoTEMPO treatment markedly restored the depolarized MMP in D-Gal induced aged H9c2 cells; (**C**) representative confocal images of chloromethyl-2′,7′-dichlorodihydrofluoroscein diacetate (DCFDA, 10-µM for 60-min incubation) loaded H9c2 cells. To monitor the ROS level, the DCFDA loaded cells were calibrated with H_2_O_2_ (100-μM). Results are given as percentage changes in the fluorescence intensities; (**D**) estimated ROS levels of all groups. Scale bar: 10-µm. Numbers of used cardiomyocytes isolated from 5–6 rats/groups were >25 per fluorescence protocols. Data were presented as Mean±SEM. Significance level accepted as * *p* < 0.05 vs. untreated H9c2 group, ^#^
*p* < 0.05 vs. D-Gal treated H9c2 group.

**Figure 4 ijms-20-03783-f004:**
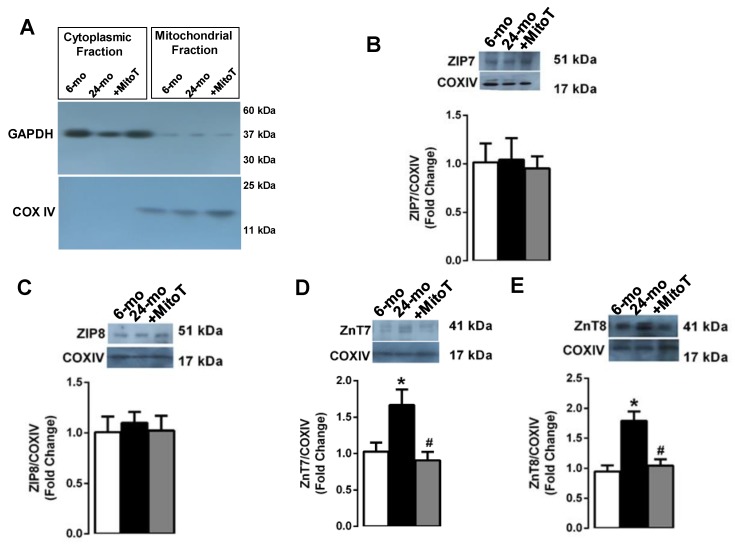
Protein expression levels of ZIP7, ZIP8, ZnT7 and ZnT8 in isolated mitochondrial fraction of left ventricular cardiomyocytes. (**A**) To validate the proper isolation of mitochondrial-cytosolic fractions, COX4 (at 17 kDa) was used as a mitochondrial marker, while GAPDH (at 37 kDa) was a cytosolic marker. MitoTEMPO treatment did not change the protein expression levels of ZIP7 (**B**) and ZIP8 (**C**), while significantly reduced the levels of ZnT7 (**D**) and ZnT8 (**E**) in the aged left ventricular cardiomyocytes. Cardiomyocytes isolated from 4–5 rats/groups were used. Data were obtained with double assays in each sample from each group for each type of measurement (numbers of rats/groups = 4–5) and were presented as Mean±SEM. Significance level accepted as **p* < 0.05 vs. 6-month old group, ^#^
*p* < 0.05 vs. 24-month old group.

**Figure 5 ijms-20-03783-f005:**
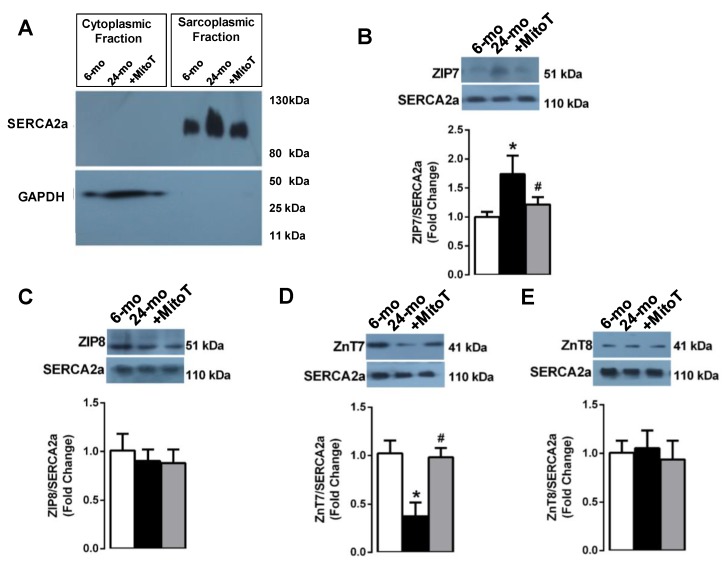
Protein expression levels of ZIP7, ZIP8, ZnT7, and ZnT8 in isolated endoplasmic reticulum fraction of the left ventricular cardiomyocytes. (**A**) To validate proper isolation of Sarco(endo)plasmic, (S(E)R), and cytosolic fractions, SERCA2a (at 110 kDa) was used as S(E)R marker, while GAPDH (at 37 kDa) was used as a cytosolic marker. Aging did not change the protein expression levels of ZIP8 (**C**) and ZnT8 (**E**) in the isolated S(E)R fraction, while there was a significant increase in ZIP7 (**B**) and decrease in ZnT7 decreased (**D**) measured in the aged cardiomyocytes. MitoTEMPO treatment preserved all these changes significantly. Data were obtained with double assays in each sample from each group for each type of measurement (numbers of rats/groups = 4–5) and were presented as Mean±SEM. Significance level accepted as * *p* < 0.05 vs. 6-month old group, ^#^
*p* < 0.05 vs. 24-month old group.

**Figure 6 ijms-20-03783-f006:**
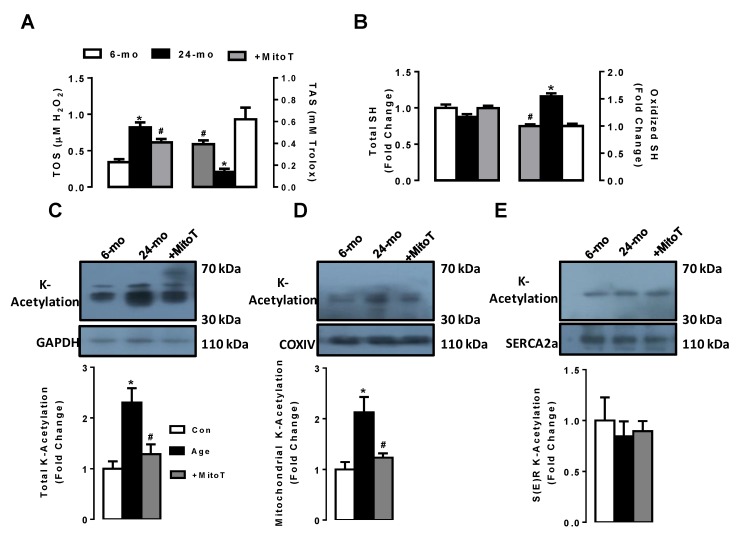
Determination of protein thiol oxidation level and protein K-Acetylation status in the aged cardiomyocytes. (**A**) total antioxidant status and total oxidant status and MitoTEMPO effects on these parameters measured in the aged cardiomyocytes; (**B**) protein thiol oxidation and MitoTEMPO effects measured in the aged cardiomyocytes; (**C**) acetylated lysine (K-Acetylation) level measured with the immune-blotting method in the aged cardiomyocytes (left), and (**D**) mitochondria-graded (middle) and (**E**) sarco(endo)plasmic-graded (right) fractions of the cardiomyocytes. Data were obtained with double assays in each sample from each group for each type of measurement (numbers of rats/groups = 4–5) and were presented as Mean ± SEM. Significance level accepted as * *p* < 0.05 vs. 6-month old group, ^#^
*p* < 0.05 vs. 24-month old group.

**Figure 7 ijms-20-03783-f007:**
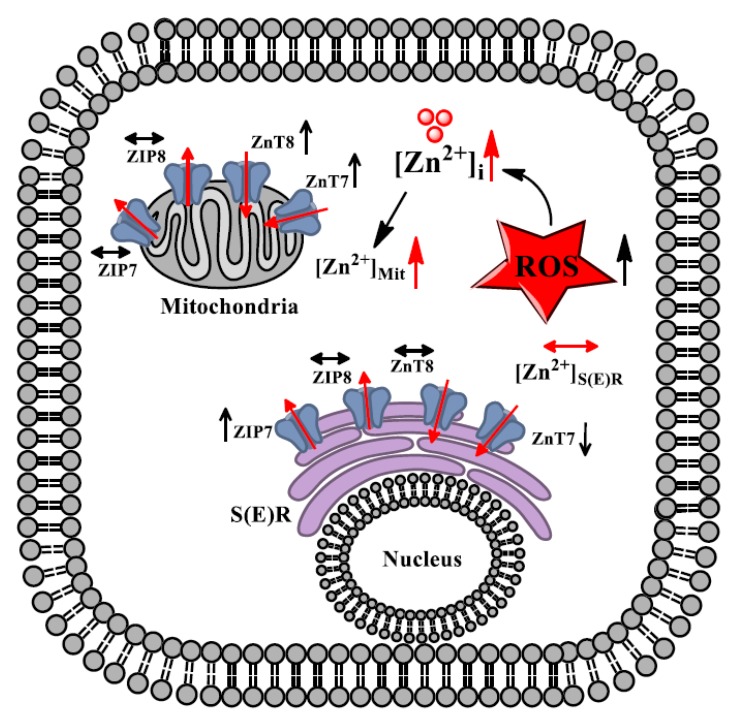
A schematic representation to demonstrate the re-distribution of [Zn^2+^]_i_ and [Zn^2+^]_Mit_ with re-distribution of Zn^2+^-transporters in the aged cardiomyocytes (24-month old rats) during the development of aging-associated cardiac dysfunction. Our data on redistribution of between cytosol, mitochondria and sarco(endo)plasmic reticulum S(E)R ([Zn^2+^]_i_, ([Zn^2+^_Mit_, ([Zn^2+^]_SER_, respectively), at most, via redistributed Zn^2+^–transporters are summarized in the aged cardiomyocytes. The close association between increased ROS and [Zn^2+^]_i_ are also considered to be an important contributor to that of insufficient cardiac function in old individuals. Here, ↑, ↓, and ↔ are representing an increase, decrease, and no change in the expression levels of Zn^2+^–transporters.

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
