# Peer review of "Mitochondria-Targeting Antioxidant Provides Cardioprotection through Regulation of Cytosolic and Mitochondrial Zn^2+^ Levels with Re-Distribution of Zn^2+^-Transporters in Aged Rat Cardiomyocytes"

_ijms, 2019, doi:10.3390/ijms20153783_

Round 1
Reviewer 1 Report
The work Mitochondria-targeting antioxidant provides cardioprotection through regulation of cytosolic and mitochondrial Zn2+ levels with re-distribution of Zn2+- transporters in aged rat cardiomyocytes has been submitted to IJMS by a group of authors with vast experience in the field and with several publications on this area. Still, although well written, some inconsistences need to be corrected and some issues need to be addressed as to be suitable for publication. The methods section is poor and needs extra work, namely better description of protocols and experiments (MitoTEMPO). Calcium currents should be evaluated also to assess the role of that ion on this story.
Abstract: although it tells a nice story with clear objectives, in the full length manuscript that clarity is not present. Moreover, never in the abstract the H9c2 cells appear and some results described belong to them and not rat adult cardiomyocytes.
Introduction: well written but with some typos and grammar inconsistencies.
Line 56: correct among observations to among other observations
Line 60: Rephrase what causes to aging-associated changes to cellular insufficient function and mitochondria are the
Line 64 correct: to expose to exposure
Line 69 correct: already knowns to already known
Line 72 to 78: since this articles relates to Zinc, I believe that Calcium is given to much info, unless authors in these models also perform calcium determinations. Or place line 85 to 87 before.
Line 94: a dot is missing
Line 83: widely discussed
The authors mention in the introduction lipid peroxidation, but never do it. Any literature available regarding it and zinc?
Methods:
The reference 24 refers to rabbits. No significant changes were performed in the protocol to rats? Rats were bought or raised in the vivarium?
Regarding the isolation procedure and cardiomycytes isolation procedure, what was the viability obtained and how many rod cells per experiments? Was percoll used for purification?
Describe thoroughly in the methods the protocol for the use of MitoTEMPO and justify concentrations (in both models the concetrations and times changed). I did not find that info anywhere on the methods only in the legend of the figures.
Regarding H9c2 cells, what were the passages used?
Isolation of cardiomyocytes is hard and requires a lot of animals, so I understand the use of H9c2 cells, but the authors need to justify why they did some experiments on the cell line and other on the isolated cells, as H9c2 cells have neonatal markers and of myoblast and not of fully developed cardiomyocytes.
(w/v) or (wt/vo)?
In point 4.9, I need further clarification whether this protocol does not have as interference GSH (a tripeptide and not a protein), the most abundant thiol in the cells.
Results: The authors mention the weight of animals and of heart and glucose, but we do not have the total N of animals used.
The authors wrote: As shown previously, there was a significant presence of insulin resistance in the old group with the measurement of OGTT and HOMO-IR index [6]. But they did not performed it? If they did, show the data.
Line 128: 4-5 h seems random and how were cells maintained~? Calcium tolerant isolated cardiomyocytes lose quality in low maintenance medium (Costa et al 2007 CRT). How was the viability after that 4-5 hours?
Line 146 H9c2
Discussion:
Rephrase: An important event is to increase [Zn2+]i about 70% with much less increase in [Ca2+]i in cardiomyocytes from the left ventricle of diabetic rats, parallel to an unbalanced oxidant-status and antioxidant capacity [46].
Line 300: conditions
Rephrase Line 305
Line 313: 30-Mo old rats? It is 24, right? Mention that is mitochondrial
Line 320: changed in similar fashion
Line 332 and 300: aging is physiological or pathological for the authors
The last two paragraphs of the discussion should be placed together and focused on the novelty of the work
Author Response
Comments and Suggestions for Authors
The work Mitochondria-targeting antioxidant provides cardioprotection through regulation of cytosolic and mitochondrial Zn2+ levels with re-distribution of Zn2+- transporters in aged rat cardiomyocytes has been submitted to IJMS by a group of authors with vast experience in the field and with several publications on this area. Still, although well written, some inconsistencies need to be corrected and some issues need to be addressed as to be suitable for publication. The methods section is poor and needs extra work, a namely better description of protocols and experiments (MitoTEMPO). Calcium currents should be evaluated also to assess the role of that ion on this story.
RESPONSES TO COMMENTS
Abstract: although it tells a nice story with clear objectives, in the full-length manuscript that clarity is not present. Moreover, never in the abstract the H9c2 cells appear and some results described belong to them and not rat adult cardiomyocytes.
- Most part of the abstract has been re-written, in order to response the comments of Reviewer and also provide the missing information (all red color sentences).
Introduction: well written but with some typos and grammar inconsistencies.
Line 56: correct among observations to among other observations
- It is changed.
Line 60: Rephrase what causes to aging-associated changes to cellular insufficient function and mitochondria are the
- It is changed.
Line 64 correct: to expose to exposure
- It is changed.
Line 69 correct: already knowns to already known
- It is changed.
Line 72 to 78: since this article relates to Zinc, I believe that Calcium is given to much info unless authors in these models also perform calcium determinations. Or place line 85 to 87 before.
- We would like to present our thanks for the very important comment. Yes, it is too much information for calcium. We deleted the lines 74-78 from the text. We also rephrased 85 and 87.
Line 94: a dot is missing
- It is changed
Line 83: widely discussed
- It is changed
The authors mention in the introduction of lipid peroxidation but never do it. Is any literature available regarding it and zinc?
- Two important references are also added (lines 56; ref. 16 and 17).
Methods:
The reference 24 refers to rabbits. No significant changes were performed in the protocol to rats? Rats were bought or raised in the vivarium?
- Actually, in the ref. 24, we used cardiomyocytes from rabbits. To isolate cardiomyocytes from rat heart, we basically used the similar isolation protocol and we mentioned slight modifications in ref. 59. Therefore, in this study, we isolated cardiomyocytes due to Ref. 59. The rats were raised in the vivarium of our faculty experimental animals house.
Regarding the isolation procedure and cardiomyocytes isolation procedure, what was the viability obtained and how many rod cells per experiments? Was personal used for purification?
As we mentioned in our previous studies (Ref. 59), we obtained %70-80 of rod-shaped cells in every heart. These cells were used for all experiments. Percoll is not used for purification.
Describe thoroughly in the methods the protocol for the use of MitoTEMPO and justify concentrations (in both models the concentrations and times changed). I did not find that info anywhere on the methods only in the legend of the figures.
- We added more information for cell isolation and the MitoTEMPO treatment of cells (lines 373-383).
Regarding H9c2 cells, what were the passages used?
- Passage number 10-15 is used for all experiments
Isolation of cardiomyocytes is hard and requires a lot of animals, so I understand the use of H9c2 cells, but the authors need to justify why they did some experiments on the cell line and other on the isolated cells, as H9c2 cells have neonatal markers and of myoblast and not of fully developed cardiomyocytes.
- We performed some experiments by using H9c2 cell lines. Not due to the hardness of freshly cell isolation or numbers of animals had to use for cell isolation, because we can only transfect cell lines with mitochondrial and ER-targeted zinc sensitive eCALWY FRET probes for measuring mitochondria and ER free Zn2+ levels. This reasoning procedure was also explained in our previous studies (Chabosseau, P, et al. 2014; Tuncay et al. 2017; Tuncay et al. 2019).
(w/v) or (wt/vo)?
- It will be w/v. It is correct
In point 4.9, I need further clarification whether this protocol does not have as interference GSH (a tripeptide and not a protein), the most abundant thiol in the cells.
We measured total thiol oxidation level to observe the oxidative status of the cardiomyocytes. Therefore, this protocol is used previously in several articles since 1996 (Turan B, et al. Cardiovasc. Res.)
Results: The authors mention the weight of animals and of heart and glucose, but we do not have the total N of animals used.
- A number of the animals is 9 to 10 per group for all presented data including cardiomyocyte isolation and biochemical analysis. These numbers are included in the first section of results (lines 106 and 107) as well as in figure legends.
The authors wrote: As shown previously, there was a significant presence of insulin resistance in the old group with the measurement of OGTT and HOMO-IR index [6]. But they did not perform it? If they did, show the data.
- We added the data related to OGTT and HOME-IR index in the first session of results (lines 111-116).
Line 128: 4-5 h seems random and how were cells maintained~? Calcium tolerant isolated cardiomyocytes lose quality in low maintenance medium (Costa et al 2007 CRT). How was the viability after that 4-5 hours?
- All experiments were performed up to 6-7 hours following to the isolation. Cells were gradually adapted to calcium with a final concentration of 1 mM. This concentration is sufficient for cellular and mechanical properties of the cells. Viability of the cells is slightly decreased due to the graded manner of adaptation (this procedure now is given in methods, lines 382-385).
Line 146 H9c2
- It is changed
Discussion:
Rephrase: An important event is to increase [Zn2+]i about 70% with much less increase in [Ca2+]i in cardiomyocytes from the left ventricle of diabetic rats, parallel to an unbalanced oxidant-status and antioxidant capacity [46].
- We would like to present our appreciation to the comment, we rephrased that sentence (lines 286-289).
Line 300: conditions
- It is changed.
Rephrase Line 305
- We are sorry for missing words, we rephrased that lines (lines 313-317).
Line 313: 30-Mo old rats? It is 24, right? Mention that is mitochondrial
- We are very sorry for this mistake. It is replaced with 24-mo old rats (line 325).
Line 332 and 300: aging is physiological or pathological for the authors
- These contradictory phrases are corrected (lines 304, 311).
The last two paragraphs of the discussion should be placed together and focused on the novelty of the work.
- We would like to present our thanks for the very important comment. They are placed together.
Reviewer 2 Report
In the paper” Mitochondria-targeting antioxidant provides cardioprotection through regulation of cytosolic and mitochondrial Zn2+ levels with re-distribution of Zn2+- transporters in aged rat cardiomyocytes” the authors investigated Zn2+-transporters and interdependence of oxidative stress and high intracellular [Zn2+]i. The cellular biology of zinc includes the storage of zinc2+ in various cellular organelles, from which zinc ions are released in a controlled way. Zinc itself is redox-inert, and most of the zinc is protein-bound with high affinity. Any increases of cytosolic free zinc ion concentrations have potent biological effects. The article contains interesting new data on regulation of cytosolic and mitochondrial Zn level. For further improvement of the manuscript, the authors should address the following specific points:
Abstract, page 1.
Line 14. It would be good to give decoding of S(E)R
Lines 15 and 18. What authors were willing to point as “Second, we…” ?
Page 2, line 44. It would be good to decode QT
Page 3. Fig.1. A very small font in the notation of the curves in Fig.1B (left).
Page 4, lines 127-128. This is the text about expression level of Zn-transporters? What does it mean Scale bar 20 µM in the line 129?
Page 4-5. Lines 153-156. Figure legend referred to Fig.2 D as estimated Zn level in mitochondria, where is figure 2D? What is indicated as 2D? Figure 2F?
Page 7. Line 216. ZIP8 – it is not B. ZnT8 – it is not C. And ZnT7 ? Do you mean ZIP7 on Fig.5B?
Page 9, line 282. Although Zn2+ itself is not a redox-inert element. This is true?
Author Response
Comments and Suggestions for Authors
In the paper” Mitochondria-targeting antioxidant provides cardioprotection through regulation of cytosolic and mitochondrial Zn2+ levels with re-distribution of Zn2+- transporters in aged rat cardiomyocytes” the authors investigated Zn2+-transporters and interdependence of oxidative stress and high intracellular [Zn2+]i. The cellular biology of zinc includes the storage of zinc2+ in various cellular organelles, from which zinc ions are released in a controlled way. Zinc itself is redox-inert, and most of the zinc is protein-bound with high affinity. Any increases of cytosolic free zinc ion concentrations have potent biological effects. The article contains interesting new data on the regulation of cytosolic and mitochondrial Zn level. For further improvement of the manuscript, the authors should address the following specific points:
Abstract, page 1.
Line 14. It would be good to give decoding of S(E)R. Lines 15 and 18. What authors were willing to point as “Second, we…”? It is comprehensively reorganized.
Due to the comments of also other Reviewers, we re-phrased and added new information into abstract as well as S(E)R is decoded (lines 13-24, 26, 29, 32).
Introduction:
Page 2, line 44. It would be good to decode QT
- It is re-phrased (line 44).
Page 3. Fig.1. A very small font in the notation of the curves in Fig.1B (left).
- It is changed.
Page 4, lines 127-128. This is the text about the expression level of Zn-transporters? What does it mean Scale bar 20 µM in line 129?
- We are sorry the repeating text information, we deleted them from figure legend. Scale bar is used for the imaged area in the confocal imaging of the cells showed with a yellow line on the right bottom of the images (lines 131-135).
Page 4-5. Lines 153-156. Figure legend referred to Fig.2 D as estimated Zn level in mitochondria, where is figure 2D? What is indicated as 2D? Figure 2F?
- Sorry for missing the labeling: There was a mistake in Fig. 2 and therefore figure legends. Right now, we inserted correct Figure 2 and provided proper figure legends.
Page 7. Line 216. ZIP8 – it is not B. ZnT8 – it is not C. And ZnT7? Do you mean ZIP7 on Fig.5B?
- They are corrected (lines 217-220).
Page 9, line 282. Although Zn2+ itself is not a redox-inert element. This is true?
- We are very sorry for the mistake in the sentence. That sentence is corrected (line 286).
Reviewer 3 Report
To:
Editorial Board
International Journal of Molecular Sciences
Title: “Mitochondria-targeting antioxidant provides cardioprotection through regulation of cytosolic and mitochondrial Zn2+ levels with re-distribution of Zn2+- transporters in aged rat cardiomyocytes”
Dear Editor,
I read this manuscript and I think that:
- The authors should include more numerical data into the results section of the abstract. Please provide.
- Please revise the English of the text due to typos.
- Indeed, the authors should take into account the role of confounding factors (age, weight, etc) on final results.
- The coupling of Zn2+ with Ca2+ metabolism had been well discussed although the research did not practically evaluate such a relationship. This is a limitation of the research paper. Please discuss such a point.
Author Response
Comments and Suggestions for Authors
To:
Editorial Board
International Journal of Molecular Science
Title: “Mitochondria-targeting antioxidant provides cardioprotection through regulation of cytosolic and mitochondrial Zn2+ levels with re-distribution of Zn2+- transporters in aged rat cardiomyocytes”
Dear Editor,
I read this manuscript and I think that:
- The authors should include more numerical data into the results section of the abstract. Please provide.
- Most part of the abstract has been re-written, in order to response the comments of Reviewer and also provide the missing information (all red color sentences).
- Please revise the English of the text due to typos.
- English of the text is revised by a professional English editing service of Ankara University.
- Indeed, the authors should take into account the role of confounding factors (age, weight, etc) on final results.
- A number of the animals is 9 to 10 per group for all presented data including cardiomyocyte isolation and biochemical analysis. These numbers are included in the first section of results (lines 106 and 107).
- We added the data related to OGTT and HOME-IR index in the first session of results (lines 111-116).
- The coupling of Zn2+ with Ca2+ metabolism had been well discussed although the research did not practically evaluate such a relationship. This is a limitation of the research paper. Please discuss such a point.
- We would like to present our appreciation to the comment, we rephrased some sentences and added 2 references required by another reviewer. Actually, since we and others had demonstrated the coupling actions of Zn2+ with Ca2+ in excitation-contraction coupling in cardiomyocytes (Tuncay et al. 2011) as well as mitochondrial function (Jang Y et al. 2007; Kamalov G, et al. 2009), we included that part in our discussion.
Round 2
Reviewer 1 Report
The authors complied to my suggestions. Some typos remained (references and scale bare 10-uM).
Suggest acceptance
Reviewer 3 Report
Dear Editor,
I read the revised version of this manuscript and I think that the authors well addressed my previous comments. The paper improved very much.